# META REINFORCEMENT LEARNING FROM OBSERVATIONAL DATA

## ABSTRACT

Pre-training is transformative in supervised learning: a large network trained with large and existing datasets can be used as an initialization when learning a new task. Such initialization speeds up convergence and leads to higher performance. In this paper, we seek to understand what the formalization for pre-training from only *existing and observational data* in Reinforcement Learning (RL) is and whether it is possible. We formulate the setting as Batch Meta Reinforcement Learning. We identify *MDP mis-identification* to be a central challenge and motivate it with theoretical analysis. Combining ideas from Batch RL and Meta RL, we propose tiMe, which learns dis**ti**llation of multiple value functions and **M**DP **e**mbeddings from only existing data. In challenging control tasks and without additional exploration on unseen MDPs, tiMe is competitive with state-of-the-art model-free RL method trained with hundreds of thousands of interactions. This work demonstrates that Meta RL from observational data is possible and we hope it will gather additional interest from the community to tackle this problem.

## 1 INTRODUCTION

Deep Reinforcement Learning algorithms still require millions of environment interactions to obtain reasonable performance, hindering their applications (Mnih et al., 2015; Lillicrap et al., 2016; Vuong et al., 2018; Fujimoto et al., 2018b; Jaderberg et al., 2018; Arulkumaran et al., 2019; Huang et al., 2019). This is due to the lack of good pre-training methods. In supervised learning, a pre-trained network significantly reduces sample complexity when learning new tasks (Zeiler & Fergus, 2013; Devlin et al., 2018; Yang et al., 2019). Meta Reinforcement Learning (RL) has been proposed as a framework for pre-training in RL (Wang et al., 2016a; Duan et al., 2016; Finn et al., 2017). However, such methods still require the collection of millions of interactions during meta-train, which means that they face the same sample complexity challenge as standard RL algorithms. In this work, we use the following definition of pre-training: the ability to use data from a set of tasks to improve performance on unseen, but related tasks.

In supervised learning, a key reason why pre-training is incredibly successful is that the dataset used for pre-training can be collected from naturally occurring large-scale processes. This removes the need to manually collect data and allows for scalable data collection, resulting in massive datasets. For example, Mahajan et al. (2018) pre-trains using existing images and their corresponding hashtags from Instagram to obtain state-of-the-art performance on ImageNet (Russakovsky et al., 2014).

In this paper, we seek to formalize pre-training in RL in a way that allows for scalable data collection. The data used for pre-training should be purely *observational* and the policies that are being optimized for should not need to interact with the environment during pre-training. To this end, we propose Batch Meta Reinforcement Learning (BMRL) as a formalization of pre-training in RL from only existing and observational data. During training, the learning algorithms only have access to a batch of existing data collected a priori from a family of Markov Decision Process (MDP). During testing, the trained policies should perform well on unseen MDPs sampled from the family.

A related setting is Batch RL (Antos et al., 2007; Lazaric et al., 2008; Lange et al., 2012), which we emphasize assumes the existing data comes from a *single* MDP. To enable scalable data collection, this assumption must be relaxed: the existing data should come from a family of *related* MDPs. Consider smart thermostats, whose goal is to maintain a specific temperature while minimizing electricity cost. Assuming Markovian dynamics, the interactions between a thermostat and

its environment can be modelled as a MDP. Data generated by a thermostat operating in a single building can be used to train Batch RL algorithms. However, if we consider the data generated by the same thermostat operating in different buildings, much more data is available. While the interactions between the same thermostat model and different buildings correspond to different MDPs, these MDPs share regularities which can support generalization, such as the physics of heat diffusion. In section 6, we further discuss the relations between BMRL and other existing formulations.

The first challenge in BMRL is the accurate inference of the unseen MDP identity. We show that existing algorithms which sample mini-batches from the existing data to perform Q-learning style updates converge to a degenerate value function, a phenomena we term *MDP mis-identification*. The second challenge is the interpolation of knowledge about seen MDPs to perform well on unseen MDPs. While Meta RL algorithms can explicitly optimize for this objective thanks to the ability to interact with the environment, we must rely on the inherent generalizability of the trained networks. To mitigate these issues, we propose tiMe, which learns from existing data to dis**ti**ll multiple value functions and **M**DP **e**mbeddings. tiMe is a flexible and scalable pipeline with inductive biases to encourage accurate MDP identity inference and rich supervision to maximize generalization. The pipeline consists of two phases. In the first phase, Batch RL algorithm is used to extract MDP-specific networks from MDP-specific data. The second phase distills the MDP-specific networks.

To summarize, our contributions are three folds: (1) Formulation of Meta RL from observational data as Batch Meta Reinforcement Learning (BMRL) (2) A simple stage-wise approach which works well on standard benchmarks in the BMRL setting (3) Most importantly, demonstration that Meta RL from only observational data is possible. We hope this work will direct the attention of the meta RL community towards this research direction.

## 2 PRELIMINARIES

### 2.1 BATCH REINFORCEMENT LEARNING

We model the environment as a Markov Decision Process (MDP), uniquely defined as a 5 element tuple $M_i = (S, A, T_i, R_i, \gamma)$ with state space $S$, action space $A$, transition function $T_i$, reward function $R_i$ and discount factor $\gamma$ (Puterman, 1994; Sutton & Barto, 1998). At each discrete timestep, the agent is in a state $s$, pick an action $a$, and arrives at the next state $s'$ and receives a reward $r$. The goal of the agent $\pi$ is to maximize the expected sum of discounted rewards $J(\pi) = \mathbb{E}_{\tau \sim \pi, M_i}[\sum_{t=0}^{\infty} \gamma^t R_i(s_{t,i}, a_{t,i}, s'_{t,i})]$ where $\tau = (s_{0,i}, a_{0,i}, r_{0,i}, s_{1,i}, a_{1,i}, r_{1,i}, \ldots)$ is a trajectory generated by using $\pi$ to interact with $M_i$. We will consider a family of MDPs, defined formally in subsection 2.3. We thus index each MDP in this family with $i$.

In Batch RL, policies are trained from scratch to solve a *single MDP* $M_i$ using existing batch of $N$ transition tuples $\mathcal{B}_i = \{(s_{t,i}, a_{t,i}, r_{t,i}, s'_{t,i}) | t = 1, \ldots, N\}$ without any further interaction with $M_i$. At test time, we use the trained policies to interact with $M_i$ to obtain an empirical estimate of its performance $J$. Batch RL optimizes for the same objective as standard RL algorithms. However, during training, the learning algorithm only has access to $\mathcal{B}_i$ and are not allowed to interact with $M_i$.

### 2.2 BATCH-CONSTRAINED Q-LEARNING

Fujimoto et al. (2018a) identifies *extrapolation error* and value function divergence as the modes of failure when modern Q-learning algorithms are applied to the Batch RL setting. Concretely, deep Q-learning algorithms approximate the expected sum of discounted reward starting from a state-action pair $E[\sum_{t=0}^{\infty} \gamma^t R(s_t, a_t, s'_t) | s_0 = s, a_0 = a]$ with a value estimate $Q(s, a)$. The estimate can be learned by sampling transition tuples from the batch and applying the temporal difference update:

$$Q(s,a) \leftarrow (1 - \alpha_t)Q(s,a) + \alpha_t(r + \gamma Q(s', \pi(s'))) \qquad \pi(s') \in \arg\max_{a \in A} Q(s', a) \quad (1)$$

The value function diverges if Q fails to accurately estimate the value of $\pi(s')$. Fujimoto et al. (2018a) introduces Batch-Constrained Q-Learning, constraining $\pi$ to select actions that are similar to actions in the batch to prevent inaccurate values estimation. Concretely, given $s'$, a generator $G$ outputs multiple candidate actions $\{a_m\}_m$. A perturbation model $\xi$ takes each state-candidate action pair as input and generates small correction term $\xi(s', a_m)$ for each candidate. The corrected

candidate action $a_m + \xi(s', a_m)$ with the highest value as estimated by a learnt $Q$ is $\pi(s')$:

$$\pi(s') = \operatorname*{arg\,max}_{a_m + \xi(s', a_m)} Q(s, a_m + \xi(s', a_m)) \qquad \{a_m = G(s', z_m)\}_m \qquad z_m \sim N(0, 1)$$

Estimation error has also been previously studied and mitigated in model-free RL algorithms (Hasselt, 2010; van Hasselt et al., 2015; Fujimoto et al., 2018b).

## 2.3 META REINFORCEMENT LEARNING

Meta RL optimizes for average return on a family of MDPs and usually assume that the MDPs in this family share $S, A, \gamma$. Each MDP is uniquely defined by a tuple $(T_i, R_i)$. A distribution $p(T_i, R_i)$ defines a distribution over MDPs. During meta-train, we train a policy by sampling MDPs from this distribution and sampling trajectories from each sampled MDP, referred to as the meta-train MDPs. During meta-test, unseen MDPs are sampled from $p(T_i, R_i)$, referred to as the meta-test MDPs. The trained policy is used to interact with the meta-test MDPs to obtain estimate of its performance. The choice of whether to update parameters (Finn et al., 2017) or to keep them fixed during meta-test (Hochreiter et al., 2001) is left to the learning algorithms, both having demonstrated prior successes.

## 2.4 MDP IDENTITY INFERENCE WITH SET NEURAL NETWORK

A Meta RL policy needs to infer the meta-test MDP identity to pick actions with high return. Rakelly et al. (2019) introduces PEARL, which uses a set neural network (Qi et al., 2016; Zaheer et al., 2017) $f$ as the MDP identity inference function. $f$ takes as input a context set $c = \{(s_k, a_k, r_k, s'_k)\}_k$ and infers the identity of a MDP in the form of distributed representation in continuous space. The parameters of $f$ is trained to minimize the error of the critic $Q$:

$$(Q(s, a, f(c)) - (r + \bar{V}(s', f(c))))^2 \tag{2}$$

where $\bar{V}$ is a learnt state value function. PEARL also adopts an amortized variational approach (Kingma & Welling, 2013; Rezende et al., 2014; Alemi et al., 2016; Kingma & Welling, 2019) to train a probabilistic $f$, which is interpreted as an approximation to the true posterior over the set of possible MDP identities given the context set.

## 3 BATCH META REINFORCEMENT LEARNING

Let $K$ be the number of meta-train MDPs, $N$ be the number of transition tuples available from each meta-train MDP, $\theta$ be the parameter of the policy, we can formulate Batch Meta Reinforcement Learning (BMRL) as an optimization problem:

$$\operatorname*{arg\,max}_{\theta} J(\theta) = \mathbb{E}_{M_i \sim p(T_i, R_i)} \left[ \mathbb{E}_{\tau \sim \pi_\theta, M_i} \left[ \sum_{t=0}^{\infty} \gamma^t R_i(s_{t,i}, a_{t,i}, s_{t',i}) \right] \right] \tag{3}$$

where the learning algorithms only have access to the batch $\mathcal{B}$ during meta-train:

$$\mathcal{B} = \cup_{i=1}^K \mathcal{B}_i \qquad \mathcal{B}_i = \{(s_{t,i}, a_{t,i}, r_{t,i}, s'_{t,i}) | t = 1, \ldots, N\} \qquad M_i \sim p(T_i, R_i)$$

We assume we know which MDP each transition in the batch was collected from. This assumption simplifies our setting and is used to devise the algorithms. To maintain the flexibility of the formalization, we do not impose restrictions on the controller that generates the batch. However, the performance of learning algorithms generally increases as the training data becomes more diverse.

**MDP identity inference challenge** To obtain high return on the unseen meta-test MDPs, the trained policies need to accurately infer their identities (Ritter et al., 2018; Gupta et al., 2018; Humplik et al., 2019). In BMRL, previously proposed solutions based on Q-learning style updates, where mini-batches are sampled from the batch to minimize TD error, converge to a degenerate solution. subsection 5.1 provides experimental result that demonstrates the phenomena. In finite MDP, this degenerate solution is the optimal value function of the MDP constructed by the relative frequencies of transitions contained in the batch. We can formalize this statement with the following proposition.

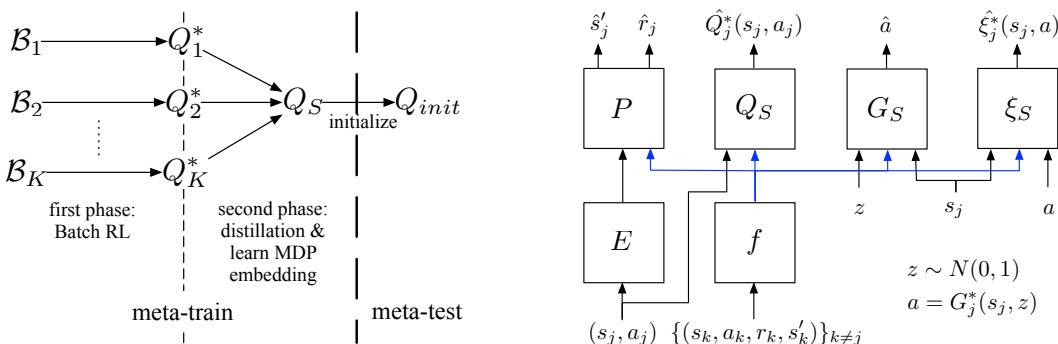

Figure 1: The above consist of two separate figures. (left) The training pipeline of tiMe in the simplest setting. (right) Architecture for the second phase when BCQ is used in the first phase.

**Proposition 1.** *Let $N(s, a, s')$ be the number of times the triple $(s, a, s')$ appears in $\mathcal{B}$ (with any reward). Performing Q-learning on finite $S$ and $A$ with all the Q-values $Q(s, a)$ initialized to $0$ and update rule (1) where $(s, a, s', r)$ is sampled uniformly at random from $\mathcal{B}$ at each step $t$, will lead the Q-values to converge to the optimal Q-value of the MDP $(S, A, \hat{T}, \hat{R}, \gamma)$ almost surely as long as $\alpha_t \geq 0$, $\sum_{t=0}^{\infty} \alpha_t = \infty$, $\sum_{t=0}^{\infty} \alpha_t^2 < \infty$, where*

$$\hat{T}(s, a, s') = \begin{cases} \mathbb{1}_{s=s'}, \text{if} \sum_{s'' \in S} N(s, a, s'') = 0 \\ \dfrac{N(s, a, s')}{\sum_{s'' \in S} N(s, a, s'')}, \text{otherwise.} \end{cases} \qquad \hat{R}(s, a, s') = \begin{cases} 0, \text{if } N(s, a, s') = 0 \\ \dfrac{\sum_{r:(s,a,s',r) \in \mathcal{B}} r}{N(s, a, s')}, \text{otherwise.} \end{cases}$$

Thus, performing Q-learning style update directly on data sampled from the batch $\mathcal{B}$ fails to find a good policy because the value function converges to the optimal value function of the *wrong* MDP. We refer this phenomena as *MDP mis-identification*. The proof is shown in subsection A.1.

**Interpolation of seen MDPs to unseen MDPs challenge** The trained policies need to generalize from the meta-train MDPs to unseen meta-test MDPs. Meta RL tackles this challenge by formulating an optimization problem that explicitly optimizes for the average return of the meta-trained policy after additional gradient steps in unseen MDPs (Finn et al., 2017; Rothfuss et al., 2018; Nichol et al., 2018). This is possible thanks to the ability to interact with the environment during meta-train. However, in the meta-train phase of BMRL, the learning algorithms do not have access to the environment. We must rely on the inherent generalizability of the trained networks to perform well on the unseen meta-test MDPs. The key challenge is therefore finding the right inductive biases in the architecture and training procedure to encourage such generalization. The need to find the right inductive biases in RL was highlighted by Botvinick et al. (2019); Zambaldi et al. (2019); Hessel et al. (2019). We note that previous works phrase the need to find inductive biases as a means to forgo generality for efficient learning. In our setting, these two goals need not be mutually exclusive.

## 4 LEARNING DISTILLATION OF VALUE FUNCTIONS AND MDP EMBEDDINGS

### 4.1 DESCRIPTION OF ARCHITECTURE AND TRAINING PROCEDURE

We propose a flexible and scalable pipeline for BMRL. Figure 1 (left) provides an overview of the pipeline in the simplest setting. Meta-train comprises of two separate phases. The first phase consists of independently training a value function $Q_i^*$ for each MDP-specific batch $\mathcal{B}_i$ using Batch RL algorithms. In the second phase, we distill the set of batch-specific value functions $\{Q_i^*\}_i$ into a super value function $Q_S$ through **supervised learning** (Hinton et al., 2015). Compared to the normal value function, a super value function takes not only a state-action pair as input, but also an inferred MDP identity, and outputs different values depending on the inferred MDP identity.

The pipeline is flexible in that any Batch RL algorithms are applicable in the first phase. Figure 1 (right) illustrates the architecture for the second phase given that the Batch RL algorithm used in

---

**Algorithm 1:** tiMe training procedure when BCQ is used in the first phase

---

**Input:** batches $\{\mathcal{B}_i\}_i$,
      $Q_S, G_S, \xi_S, f, E, P$ parameterized
      jointly by $\theta$

1   $Q_i^*, G_i^*, \xi_i^* \leftarrow BCQ(\mathcal{B}_i) \quad \forall i$
2   Randomly choose $\mathcal{B}_j$ out of $\{\mathcal{B}_i\}_i$
3   Sample a transition $(s_j, a_j, s_j', r_j)$ from $\mathcal{B}_j$
4   Sample context $\{(s_k, a_k, s_k', r_k)\}_{k \neq j}$ from $\mathcal{B}_j$
5   Infer MDP identity:
      $\hat{M} \leftarrow f(\{(s_k, a_k, s_k', r_k)\}_k)$
6   Predict $s', r$: $\hat{s}', \hat{r} \leftarrow P(E(s_j, a_j), \hat{M})$

7   Predict state-action value:
      $\hat{Q}_j^* \leftarrow Q_S(s_j, a_j, \hat{M})$
8   $z \sim N(0, 1)$
9   Predict candidate action: $\hat{a} \leftarrow G_S(s_j, z, \hat{M})$
10   Obtain ground truth candidate action:
      $a \leftarrow G_j^*(s_j, z)$
11   Predict correction factor: $\hat{\xi}_j^* \leftarrow \xi_S(s_j, a)$
12   $L \leftarrow \|\hat{s}' - s'\|_2^2 + (\hat{r} - r)^2 + (\hat{Q}_j^* - Q_j^*(s_j, a_j))^2 + \|\hat{a} - a\|_2^2 + \|\hat{\xi}_j^* - \xi_j(s_j, a)\|_2^2$
13   $\theta \leftarrow \theta - \nabla_\theta L$

---

the first phase is Batch Constrained Q (BCQ) Learning. As described in subsection 2.2, BCQ maintains three separate components, a learnt value function $Q$, a candidate action generator $G$ and a perturbation model $\xi$. Therefore, the output of the first phase consists of 3 sets $\{Q_i^*\}_i, \{G_i^*\}_i, \{\xi_i^*\}_i$.

The second phase distills $\{Q_i^*\}_i, \{G_i^*\}_i, \{\xi_i^*\}_i$ to $Q_S, G_S, \xi_S$ respectively. The distillation of $G$ and $\xi$ is necessary to pick actions that lead to high return because each learnt value function $Q_i^*$ only provides reliable estimates for actions generated by $G_i^*$ and $\xi_i^*$, a consequence of the training procedure of BCQ. In addition to $Q_S, G_S, \xi_S$, the architecture consists of 3 other networks, $f, P, E$. $f$ takes as input a context $\{(s_k, a_k, r_k, s_k')\}_k$ and outputs a distributed representation of the MDP identity in a fixed-dimension continuous space. The output of $f$ is an input to $Q_S, G_S, \xi_S, P$. $E$ and $P$ predicts $s_j', r_j$ given a state-action pair $(s_j, a_j)$. $P$ has low capacity while the other networks are relatively large. During the second phase, all networks are jointly trained end-to-end by the regression losses of predicting $s', r$ and distilling $\{Q_i^*\}_i, \{G_i^*\}_i, \{\xi_i^*\}_i$. This is illustrated in details in Algorithm 1.

During meta-test, $f$ is used to infer the identity of the meta-test MDP as a fixed-dimension continuous vector. The super functions $Q_S, G_S, \xi_S$ are used to pick actions in the meta-test MDPs, using the same procedure as BCQ (subsection 2.2). The super functions also take as input the inferred MDP identity.

The key idea behind the approach is a simple stage-wise approach for the Meta RL from observational data problem. In the second phase, we distill many policies into one for related tasks by jointly learning dis**ti**llation of value functions and **M**DP **e**mbeddings. We therefore name the approach tiMe. MDP embeddings refer to the ability to infer the identity of a MDP in the form of distributed representation in continuous space given a context.

### 4.2 BENEFITS OF THE PROPOSED PIPELINE

**Inductive biases to encourage accurate MDP identification** The first inductive bias is the relationship between $f$ and $Q_S$. They collectively receive as input state-action pair $(s_j, a_j)$ and context $\{(s_k, a_k, r_k, s_k')\}_k$ and regress to target value $Q_j^*(s_j, a_j)$. The target for each state-action pair can take on the values within the set $\{Q_i^*(s_j, a_j)\}_i$. Similar state-action pairs can have very different regression targets if they correspond to different meta-train MDPs. The context is the only information in the input to $f$ and $Q_S$ that correlates with which $Q^*(s_j, a_j)$ out of the set $\{Q_i^*(s_j, a_j)\}_i$ $f$ and $Q_S$ should regress to. Thus, $f$ and $Q_S$ must learn to interpret the context to predict the correct value for $(s_j, a_j)$. The second inductive bias is the auxiliary task of predicting $s_j', r_j$. A key design choice is that the network $P$ which takes as input $E(s_j, a_j)$ and $f(\{(s_k, a_k, r_k, s_k')\}_k)$ and predicts $s_j', r_j$ has low capacity. As such, the output of $f$ must contain meaningful semantic information such that a small network can use it to reconstruct the MDP. This is to prevent the degenerate scenario where $f$ learns to copy its input as its output. To summarize, these two explicit biases in the architecture and training procedure encourage $f$ to accurately infer the MDP identity given the context.

**Richness and stability of supervision** Previous approaches update $f$ to minimize the critic loss (subsection 2.4). It is well-known that RL provides sparse training signal. This signal can also cause

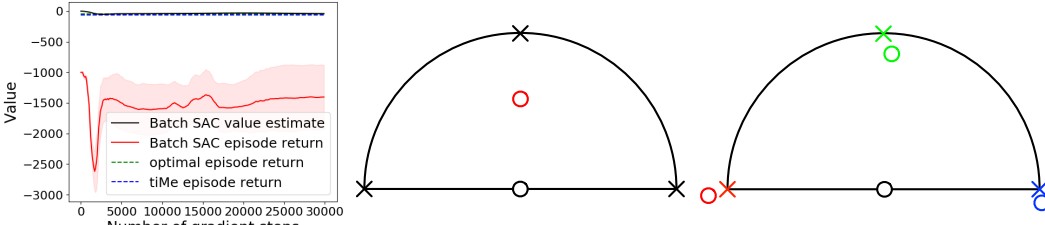

Figure 2: All three figures are from the 3 meta-train MDPs scenario. (left) Performance of Batch SAC in one meta-test MDP and the learnt value estimates of initial state-action pairs. The estimates do not diverge but are significantly higher than the actual returns, demonstrating the *MDP mis-identification* phenomena. In contrast, tiMe's performance is close to optimal. (middle) The behavior of the Batch SAC agent. The dark circle, red circle and dark crosses indicates the agent's starting locations, final location and the 3 meta-train goal locations. The agent fails to find good actions during meta-test and navigates to the location closest to the 3 meta-train goals. (right) The behavior of an agent trained with tiMe. The crosses and circles indicate each meta-test MDP goal location and the agent's corresponding final location after evaluation in the corresponding meta-test MDP. The agent trained with tiMe finds near-optimal actions in all 3 meta-test MDPs.

instability since the target values in the critic loss change over time. In contrast, our pipeline provides training signal for $f$ that is both rich and stable. It is rich because $f$ is trained to infer a representation of the MDP identity that can be used for multiple downstream tasks, such as predicting $s', r$. This encourages general-purpose learnt representation and supports generalization. The training signal is also stable since the regression targets are fixed during the second phase of tiMe.

**Scalability** The pipeline is scalable in that an arbitrary amount of purely observational data can be used in the first phase so long as computational constraints permit. The data can also be heterogeneous in the sense that they do not need to contain only trajectories with high return. In the experimental section, we demonstrate the benefit of the approach when the data contains trajectories of varying qualities, some of which were generated by random policies. The extraction of the batch-specific networks, such as the batch-specific value functions $\{Q_i^*\}_i$, from the MDP-specific batches can be trivially parallelized and scales gracefully as the number of meta-train MDPs increases.

## 5 EXPERIMENTAL RESULTS

Our experiments have two main goals: (1) Demonstration of the *MDP mis-identification* phenomena and tiMe's ability to effectively mitigate it. (2) Demonstration of the scalability of the tiMe pipeline to challenging continuous control tasks and generalization to unseen MDPs.

In all experiments, the MDP-specific batch $\mathcal{B}_i$ is the replay buffer when training Soft Actor Critic (SAC) (Haarnoja et al. (2018a;b)) in the MDP $M_i$ for a fixed number of environment interactions. Thus, the batch $\mathcal{B}_i$ contains transitions with varying reward magnitude, some of which were generated by random and poorly performing policies. While our problem formulation BMRL and the pipeline tiMe allow for varying both the transition and reward functions within the family of MDPs, we consider the case of changing reward function in the experiments and leave changing transition function to future work. Thus, for the auxiliary prediction task in the second phase of the pipeline, $P$ only predicts $r$ and not $s'$.

### 5.1 TOY EXPERIMENTS

This section illustrates *MDP mis-identification* as the failure mode of existing Batch RL algorithms in BMRL. The toy setting allows for easy interpretability of the trained agents' behavior. We also show that in the standard Batch RL setting, the Batch RL algorithm tested finds a near-optimal policy. This means the failure of existing Batch RL algorithm in BMRL is not because of the previously identified extrapolation issue when learning from existing data (Fujimoto et al., 2018a).

**Environment Description** In this environment, the agent needs to navigate on a 2d-plane to a goal location. The agent is a point mass whose starting location is at the origin $(0,0)$. Each goal location is a point on a semi-circle centered at the origin with radius of 10 units. At each discrete timestep, the agent receives as input its current location $(x, y)$, takes an action indicating the change in its position $(\Delta x, \Delta y)$, transitions to a new position $(x + \Delta x, y + \Delta y)$ and receives a reward. The reward is the negative distance between the agent's current location and the goal location. The agent does not receive the goal location as input and $|\Delta x| \leq 1, |\Delta y| \leq 1$. Since the MDP transition function is fixed, each goal location uniquely defines a MDP. The distribution over MDPs is defined by the distribution over goal locations, which corresponds to a distribution over reward functions.

**Batch SAC** We modify SAC to learn from the batch by initializing the replay buffer with existing transitions. Otherwise, training stays the same. We test Batch SAC on a simple setting where there is only one meta-train MDP and one meta-test MDP which share the same goal location. This is the standard Batch RL setting and is a special case of BMRL. Batch SAC finds a near-optimal policy.

**Three meta-train and meta-test MDPs** This experiment has 3 meta-train MDPs with different goal locations. The goals divide the semi-circle into two segments of equal length. There are three meta-test MDPs whose goal locations **coincides** with the goal locations of the meta-train MDPs. This setting only tests the ability of the trained policies to correctly identify the meta-test MDPs and do not pose the challenge of generalization to unseen MDPs. Batch SAC was trained by combining the transitions from the 3 meta-train MDPs into a single replay buffer and sampling transitions from this buffer to perform gradient updates. Otherwise, training stays the same as SAC. Figure 2 (left, middle) illustrates that Batch SAC fails to learn a reasonable policy because of the *MDP misidentification* phenomena.

**Batch SAC with task inference function** We also tried adding to the architecture of Batch SAC the probabilistic MDP identity inference function as described in subsection 2.4. This is the equivalent of adapting PEARL (Rakelly et al., 2019) to work in the BMRL setting. This approach fails to train a policy that performs well on all 3 meta-test MDPs. We note that the off-policy meta RL setting that the PEARL paper considers and the BMRL setting we consider are solving different problems. We do not argue that one is easier than the other.

**Performance of tiMe** Since Batch SAC can extract the optimal value function out of the batch in the single meta-train MDP case, we use it as the Batch RL algorithm in the first phase of the tiMe pipeline. The architecture in the second phase thus consists of $E, P, f$ and $Q_S$. To pick an action, we randomly sample multiple actions and choose the action with the highest value as estimated by $Q_S$. This method is termed random shooting (Chua et al., 2018). As illustrated in Figure 2 (left, right), tiMe can identify the identities of the three meta-test MDPs and pick near-optimal actions.

## 5.2 MUJOCO EXPERIMENTS

**Environment Description** This section illustrates the test of tiMe in challenging continuous control robotic locomotion tasks. Each task requires the application of control action to a simulated robot so that it moves with a particular velocity in the direction of its initial heading. Formally, the MDP within each MDP family share $S, A, T, \gamma$ and only differs in $R$ where $R$ is defined to be:

$$R(s, a, s') = \text{alive\_bonus} - \alpha|\text{current\_velocity} - \text{target\_velocity}| - \beta||a||_2$$

where $\alpha$ and $\beta$ are positive constant. A one-to-one correspondence exists between a MDP within the family and a target velocity. Defining a family of MDP is equivalent to picking an interval of possible target velocity. This setting is instantiated on two types of simulated robots, hopper and halfcheetah, illustrated in Figure 3. Experiments are performed inside the Mujoco simulator (Todorov et al., 2012). The setting was first proposed by Finn et al. (2017).

**Zero-shot meta-test** During testing, in contrast to prior works, we do not update the parameters of the trained networks, as is done in gradient-based meta RL, or allow for an initial exploratory phase where episode returns do not count towards the final meta-test performance, as is done in off-policy meta RL (Rakelly et al., 2019). This allows for testing the inherent generalizability of the trained networks without confounding factors. The meta-test MDPs are chosen such that they are unseen during meta-train, i.e. none of the transitions used during meta-train was sampled from any of the meta-test MDPs. At the beginning of each meta-test episode, the inferred MDP identity is initialized to a zero vector. Subsequent transitions collected during the episode is used as the

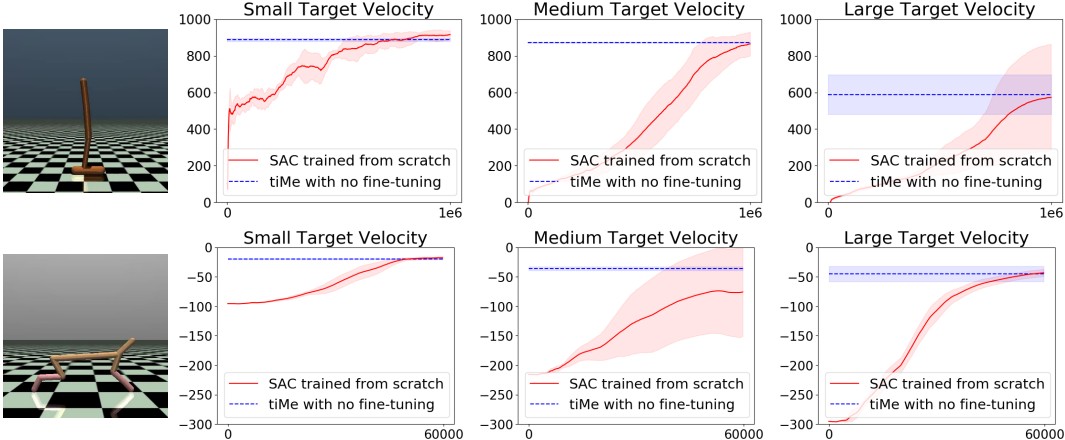

Figure 3: The leftmost column illustrates hopper and halfcheetah. The remaining columns indicate the performance of SAC trained from scratch versus tiMe for different **unseen** MDPs during **zero-shot** meta-test. Because of these two properties, we highlight the difficult nature of obtaining high performance in this setting. The second to forth columns correspond to small, medium, and large target velocities respectively. The first and second row indicates performance on hopper and halfcheetah respectively. The x-axis indicates SAC's number of environment interactions. The y-axis indicates the average episode return. The final performance of SAC is close-to-optimal in all plots in the sense that running SAC for many more timesteps will not increase its performance significantly.

context. The meta-test MDPs are also chosen to provide wide coverage over the support of the MDP distribution. This is to test whether our approach generalizes to a variety of meta-test MDPs, or simply overfit to a small region inside the support.

**Meta-train conditions** The target velocities of the meta-train MDPs divide the target velocity interval into equal segments. This removes the bias of sampling meta-train MDPs when evaluating performance. The target velocity intervals, episode length, and number of meta-train MDPs for hopper and halfcheetah are $[0.0, 2.35]$ and $[0.0, 1.5]$, 1000 and 200, 16 and 29 respectively. For hopper and halfcheetah, each meta-train MDP has one million and sixty transitions respectively.

**Performance analysis** Figure 3 illustrates tiMe's performance on **unseen** meta-test MDPs. tiMe is competitive with the state-of-the-art model-free RL methods trained from scratch for one million and sixty thousands environment interactions in hopper and halfcheetah respectively. We perform experiments on halfcheetah with an episode length 200 because of computational constraints. Previous Meta RL works also use an episode length of 200 (Rakelly et al., 2019). The same network trained with tiMe also performs well in a variety of different meta-test MDPs, demonstrating that it does not over-fit to one particular meta-train MDP. We compare with SAC to demonstrate BMRL is a promising research direction. We do not include other Meta RL algorithms as baseline because they would require interacting with the environment during meta-train and thus, is not solving for the problem that BMRL poses. We tried removing $G_S, \xi_S$ from the architecture in Figure 1 and picked action with Cross Entropy Method (Rubinstein & Kroese, 2004), but that lead to poor performance because $Q_S$ over-estimates the values of actions not generated by $G_S, \xi_S$.

**Limitations** Our approach assumes that the transitions in the batch contain good enough transitions to learn a good policy in the batch RL setting. However, we note that the data in the batch contain data of varying qualities, some of which were generated by poorly performing policies. Also, our approach has only been demonstrated to work on tasks where reset are not crucial for exploration needed for task inference, e.g. sparse reward setting. We leave this venue for future work.

## 6 RELATED WORKS

**Supervised Learning and Imitation Learning** The main differences between Batch (Meta) RL and supervised learning are: actions have long-term consequences and the actions in the batch are not assumed to be optimal. If they are optimal in the sense that they were collected from an expert, Batch RL reduces to Imitation Learning (Abbeel & Ng, 2004; Ho & Ermon, 2016). In fact, Fujimoto et al. (2018a) demonstrates that Batch RL generalizes Imitation Learning in discrete MDPs.

**Meta RL** Equation 3 is the same objective that existing Meta RL algorithms optimize for (Wang et al. (2016b); Finn et al. (2017)). We could have formulated our experimental setting as a Partially Observable MDP, but we chose to formulate it as Batch Meta Reinforcement Learning to ensure consistency with literature that inspires this paper. The main difference between Meta RL and our formulation is access to the environment during training. Meta RL algorithms sample transitions from the environment during meta-train. We only have access to existing data during meta-train.

**Context Inference** Zintgraf et al. (2019) and Rakelly et al. (2019) propose learning inference modules that infer the MDP identity. Their procedures sample transitions from the MDP during meta-train, which differs from our motivation of learning from only existing data. Killian et al. (2017) infers the MDP's "hidden parameters", inputs the parameters to a learnt transition function to generates synthetic data and train a policy from the synthetic data. Such model-based approaches are still outperformed by the best model-free methods (Wang et al. (2019)), which our method is based on.

**Batch RL** Fujimoto et al. (2018a) and Agarwal et al. (2019) demonstrate that good policies can be learnt entirely from existing data in modern RL benchmarks. Our work extends their approaches to train policies from data generated by a family of MDPs. Li et al. (2004) selects transitions from the batch based on an importance measure. They assume that for state-action pair in the batch, their value under the optimal value function can be easily computed. We do not make such assumption.

**Factored MDPs** In discrete MDP, the number of possible states increases exponentialy in the number of dimension. Kearns & Koller (2000) tackles this problem by assuming each dimension in the next state is conditionally dependent on only a subset of the dimensions in the current state. In contrast, our method makes no such assumption and applies to both discrete and continuous settings.

**Joint MDP** The family of MDPs can be seen as a joint MDP with additional information in the state which differentiates states between the different MDPs (Parisotto et al., 2015). Sampling an initial state from the joint MDP is equivalent to sampling a MDP from the family of MDPs. However, without prior knowledge, it is unclear how to set the value of the additional information to supports generalization from the meta-train MDPs to the meta-test MDPs. In fact, the additional information in our approach is the transitions from the MDP and the network learns to infer MDP identity.

## 7 CONCLUSION

We propose a new formalization of pre-training in RL as Batch Meta Reinforcement Learning (BMRL). BMRL differs from Batch RL in that the existing data comes from a family of related MDPs and thus enables scalable data collection. BMRL also differs from Meta RL in that no environment interaction happens during meta-train. We identified two main challenges in BMRL: MDP identity inference and generalization to unseen MDPs. To tackle these challenges, we propose tiMe, a flexible and scalable training pipeline which jointly learn distillation of value functions and MDP embeddings. Experimentally, we demonstrate that tiMe obtains performance competitive with those obtained by state-of-the-art model-free RL methods on unseen MDPs.

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

# A  APPENDIX

## A.1  MDP MIS-IDENTIFICATION CONVERGENCE PROOF

*Statement*: Performing Q-learning on finite $S$ and $A$ with all the $Q$-values $Q(s, a)$ initialized to $0$ and update rule (1) where $(s, a, s', r)$ is sampled uniformly at random from $\mathcal{B}$ at each step $t$, will lead the $Q$-values to converge to the optimal $Q$-value of the MDP $(S, A, \hat{T}, \hat{R}, \gamma)$ almost surely as long as $\alpha_t \geq 0$, $\sum_{t=0}^{\infty} \alpha_t = \infty$, $\sum_{t=0}^{\infty} \alpha_t^2 < \infty$, where

$$
\hat{T}(s, a, s') = \begin{cases} \dfrac{N(s, a, s')}{\sum_{s'' \in S} N(s, a, s'')}, & \text{if } \sum_{s'' \in S} N(s, a, s'') > 0 \\[2ex] \mathbb{1}_{s=s'}, & \text{otherwise.} \end{cases}
$$

$$
\hat{R}(s, a, s') = \begin{cases} \dfrac{\sum_{r:(s,a,s',r)\in\mathcal{B}} r}{N(s, a, s')}, & \text{if } N(s, a, s') > 0 \\[2ex] 0, & \text{otherwise.} \end{cases}
$$

*Proof.* First note that for any $(s, a) \in S \times A$ such $\sum_{s'' \in S} N(s, a, s'') = 0$, the initial $Q(s, a)$ is already optimal and will never be updated; for all other $(s, a) \in S \times A$ and any $s' \in S$, we have

$$
\hat{T}(s, a, s') = \Pr_{(s_0, a_0, s_0', r_0) \sim \mathcal{B}} (s_0' = s' | s_0 = s, a_0 = a),
$$

$$
\hat{R}(s, a, s') = \mathbb{E}_{(s_0, a_0, s_0', r_0) \sim \mathcal{B}} [r_0 | s_0 = s, a_0 = a, s_0' = s'],
$$

and with probability 1,

$$
\sum_{t=0}^{\infty} \alpha_t \mathbb{1}_{\{Q(s, a) \text{ is updated at round } t\}} = \infty
$$

$$
\sum_{t=0}^{\infty} \alpha_t^2 \mathbb{1}_{\{Q(s, a) \text{ is updated at round } t\}} < \infty
$$

Then convergence follows from the same argument for the convergence of $Q$-learning (Watkins & Dayan, 1992) □

## B  HYPER-PARAMETERS

The small, medium and large target velocity in hopper corresponds to $0.2, 1.1, 2.0$. The small, medium and large velocity in halfcheetah corresponds to $0.475, 1.075, 1.475$. The learning rate is $3e4$ and the Adam optimizer is used in all experiment. All neural networks used are feed-forward network. All experiments are performed on machines with up to $48$ CPU cores and $4$ Nvidia GPU.

All experiments are performed in Python 3.7, mujoco-py 2.0.2.5 running on top of mujoco 200. All neural network operations are in Pytorch 1.2.

In the toy experiment, $Q_S$ consists of 2 hidden layers, each of size 256. The inferred MDP size is 32. The context size is 1. $f$ consists of 1 hidden layers of size 256. $E$ consists of 1 hidden layers of size 256 and outputs 256 values. Same goes for $P$. Random shooting was performed with 100 random actions at each iterations.

In hopper, the size of the inferred MDP is 8. The context size is 1. $f$ consists of 3 hidden layers, each of size 256. $E$ consists of 4 hidden layers, each of size 16, and outputs a vector of size 8. $B$ consists of 2 hidden layers, each of size 4. $Q_S$ consists of 8 hidden layers, each of size 128. The training mini-batch size is 32. When BCQ is ran to extract the value function out of the batch, the same hyper-parameters as found in the official implementation are used `https://github.com/sfujim/BCQ`, except the learning rate is lowered from 0.003 to 0.0003. The $alpha$ and $beta$ in the reward function definition are 1.0 and 0.001. The $alive\_bonus$ is 1.0.

In halfcheetah, the size of the inferred MDP is 64. The context size is 1. $f$ consists of 7 hidden layers, each of size 512. $E$ consists of 7 hidden layers, each of size 512, and outputs a vector of size 64. $B$ consists of 1 hidden layers, each of size 64. $Q_S$ consists of 8 hidden layers, each of size 512. The training mini-batch size is 64. $G_S$ consists of 7 hidden layers, each of size 750. $\xi_S$ consists of 7 hidden layers, each of size 400. When BCQ is ran to extract the value function out of the batch, unless otherwise mentioned, the same hyper-parameters as found in the official implementation are used `https://github.com/sfujim/BCQ`. The learning rate is lowered from 0.003 to 0.0003. The perturbation model has 2 hidden layers, of size $400, 300$. The critic also has 2 hidden layers, of size $400, 300$. The $alpha$ and $beta$ in the reward function definition are 1.0 and 0.05. The $alive\_bonus$ is 0.0.

In both hopper and halfcheetah, except for the super Q function loss, the terms in the loss $L$ in Algorithm 1 are scaled so that they have the same magnitude as the super Q function loss. Graphs for the mujoco experiments are generated by smoothing over the last 100 evaluation datapoints.

The performance on Mujoco was averaged over 5 seeds $0-4$. The hyper-parameters for SAC are the same as those found in the Pytorch public implementation `https://github.com/vitchyr/rlkit`. The standard deviations are averaged over 5000 timesteps during evaluation. This corresponds to 5 episodes in halfcheetah because there is no terminal state termination in halfcheetah and variable number of episodes in hopper because there is terminal state termination.

