# OpenReview forum: "Pre-training as Batch Meta Reinforcement Learning with tiMe "
_ICLR.cc/2020/Conference — Reject_

### Official Review · AnonReviewer3 · 2019-10-23
**Official Blind Review #3**

**Rating:** 3

**Review:**

The paper studies batch meta learning, i.e. the problem of using a fixed experience from past tasks to learn a policy which can quickly adapt to a new related task. The proposed method combines the techniques of Fujimoto et al. (2018) for stabilizing batch off-policy learning with ideas from Rakelly et al. (2019) for learning a set invariant task embedding using task-specific datasets of transitions. They learn task-specific Q-values which are then distilled into a new Q function which is conditioned on the task embedding instead of the task ID. The embedding is further shaped using a next-state prediction auxiliary loss. The algorithmic ideas feel a bit too incremental and the experimental evaluation could be stronger--I'd recommend trying the method on more complicated environments and including ablation studies.

Specific comments:
1. I disagree that the cheetah and hopper environments are "challenging"--they're one of the simplest MuJoCo environments.
2. The problem of adapting to run at a specific speed when the meta-learner observes the dense rewards is actually not a meta learning problem because the meta learner can uniquely identify the target speed from a single transition. This is because the current speed is part of the observation, and so given the value of the dense reward at this state, it is simple to calculate the target speed. Hence these environments are effectively the same as directly giving the agent the target speed as an input. Given this interpretation, I'm not sure what is "meta" about this environment. The problem then reduces to the question of whether the agent can generalize from the 16 or 29 training tasks. That this should be the case is not surprising considering the one-dimensional nature of the task space.
3. It would also be useful to see some ablations. For example, is the auxiliary prediction task necessary? Would it be possible to side step the distillation process and directly learn Q_S from the buffer as done e.g. in Rakelly et al. (2019)? Could you show some data that the corrections from Fujimoto et al. (2018) are important in the batch setting?

------------------------------------------------------------------------------------------------------------
Thanks for your comments. I still think this is too incremental, and my concerns regarding the environments and using the dense reward as a feature which identifies the task haven't changed and so I'm keeping my score as is.

**Experience Assessment:**

I have published one or two papers in this area.

**Review Assessment: Checking Correctness Of Derivations And Theory:**

N/A

**Review Assessment: Checking Correctness Of Experiments:**

I assessed the sensibility of the experiments.

**Review Assessment: Thoroughness In Paper Reading:**

I made a quick assessment of this paper.

---

> ### Author Response · Authors · 2019-11-06
> **Reply**
>
> Thank you for your comments. We will revise the paper to make the exposition clearer based on your comments.
>
> Please let us know if you are happy with our answers.
>
> Please find our reply to the specific points you raised below:
>
> 1. I disagree that the cheetah and hopper environments are "challenging"--they're one of the simplest MuJoCo environments.
>
> We are arguing that the experimental setting is challenging because the algorithm is tested on (1) unseen MDPs during (2) zero-shot meta-test. It is the combination of these two conditions that make the setting challenging.
>
> 2. ...Given this interpretation, I'm not sure what is "meta" about this environment...
>
> This type of environment where the agent needs to hit a target velocity is one of the standard benchmarks in the meta RL community. To list a few prior published works that demonstrate the effectiveness of their approaches in this type of environment:
>
> https://arxiv.org/abs/1703.03400 (MAML - ICML 2017)
> http://proceedings.mlr.press/v97/rakelly19a/rakelly19a.pdf (PEARL - ICML 2019)
> http://proceedings.mlr.press/v97/liu19g/liu19g.pdf (TMAML - ICML 2019)
>
> 3.
>
> Is the auxiliary prediction task necessary?
>
> We agree that this is a relevant ablation and will try to run it before the update period is over.
>
> Would it be possible to side step the distillation process and directly learn Q_S from the buffer as done e.g. in Rakelly et al. (2019)?
>
> We tried the approach proposed by Rakelly et al. (2019) and it did not successfully identify the identity of the MDP during meta-test, leading to poor test-time performance. This is mentioned in section 5.1, at the end of the paragraph  “three meta-train and meta-test MDPs”. We will make this point more prominent in the revised paper.
>
> Could you show some data that the corrections from Fujimoto et al. (2018) are important in the batch setting?
>
> Figure 7 in Appendix D.1 in the Fujimoto paper provides the answer to your question. Since our approach is about batch meta reinforcement learning and not about batch reinforcement learning, we use the algorithm as proposed by  Fujimoto et al. (2018) without changes.

---

### Official Review · AnonReviewer2 · 2019-10-23
**Official Blind Review #2**

**Rating:** 3

**Review:**

This paper studies the meta-RL problem in the off-policy, batch learning setting. Batch-RL is the setting in which a policy is learned entirely offline, that is, without interaction with the environment and given only trajectories collected by some policy. Compared to RL, Meta-RL involves the additional challenge of task-inference; the goal of Meta-RL is to train a policy that can generalize to a distribution of tasks (i.e. a distribution of MDPs), without actually being given a description of the task (unlike contextual multi-task policy learning). A simple approach for solving the meta-RL problem thus might be to first perform task inference by encoding data from some task into a task description, and then condition a contextual policy on this task description.

The authors state that the straight-forward application of such an idea in the Batch-RL setting fails due to an issue they term the MDP “misidentification” problem, wherein having multiple tasks in a single batch results in confusion between tasks. This issue really only arises in the setting where task inference is jointly learned with the multi-tasking contextual policy. Thus, they propose a stage-wise algorithm wherein first 1) the N tasks in some dataset of trajectories are learned by N separate policies, and then 2) these N separate policies are distilled into a single master policy, wherein the subpolicy enacted by the master policy is modulated by some sort of task description. The distillation procedure thus involves task description (i.e. mapping from task data to a task embedding) and supervised learning (i.e. imitation) of each sub policy when conditioned on respective task embedding.

* Pros
	* Demonstrates the effectiveness of a straightforward stage-wise approach for off-policy meta-RL (albeit without comparison to alternatives)
	* The method seems to perform well on the tasks considered, approaching the level of performance of SOTA model-free algorithms.
	* The approach is general insofar as it could presumably be used with other batch-RL methods (for first stage of algorithm), or even non-batch RL methods. In this sense, the main contribution might be seen as an approach for distilling multiple policies into a single policy in a way that allows for interpolating between them.

* Cons
	* Technical novelty: while they address a problem that has not yet seen much attention, their solution is combination existing solutions. I say this because I am not fully convinced that the more novel aspects of their distillation procedure (i.e. the auxiliary task) are absolutely necessary. I would be willing to change my stance on this, given evidence.
		* Ablations of i.e. the auxiliary task would help to clarify this. In all tasks considered, the transition function T does not change. Therefore, the function composite P(E(s_j, a_j)) should actually not learn any task-specific information when predicting s’_j. If the method works without the auxiliary loss, it is very similar to a stagewise version of PEARL, with Batch-RL.
	* Limited comparisons to relevant alternatives, simple baselines.	 Do not compare to anything other than SAC.
	* Only tasks considered are pointmass, hopper, and half-cheetah; other work (i.e. PEARL) has also been demonstrated on Ant in Mujoco.
	* Argues that it is more stable than variants that involve interaction with the environment / use critic loss for MDP identification, which is a somewhat unfair since in their case 1) they assume they have data that is good enough to learn a good policy in the batch-RL setting, and 2) the stability is by virtue of the fact that they do things in a stage-wise manner because batch-RL works, i.e. they are doing supervised learning without having to bootstrap.
	* Not sure if the use of term “pre-train” is appropriate, insofar as the test tasks are assumed to come from the same distribution as training tasks. It seems to be more about Batch-Meta-RL.
	* Not much attention given to attempts to solve the MDP mis-identification problem with simple solutions like giving side-information at training time (i.e. task-ID).

* Would be helpful to see:
	* More baselines
		* A simple task classifier. This would basically do a nearest neighbor lookup over the training tasks (given a test task), but this might perform well under the reward function.
		* Comparison to something like PEARL, or some method that does involve interaction with the environment -> this would help shed light on whether interaction is necessary for task inference in the the tasks they consider, and if so, such methods would in some sense be oracles.
	* Ablations
		* Remove auxiliary forward prediction loss: if it works without auxiliary loss, this is very similar to a stagewise version of PEARL, with Batch-RL.
		* How much data per task is needed?
	* Harder tasks, where the method can’t approach SAC (as one would expect for sufficiently challenging tasks)
	* Experiment where they study zero-shot generalization by considering disjoint parts of task space
		* Claim zero-shot generalization but in this case they study tasks where resets are not crucial for exploration needed for task inference

* Minor comments:
	* I found the discussion about inductive bias in RL at the end of section 3 (last few sentences of last paragraph) to be a bit vague.

I've given a weak reject mainly because 1) auxiliary loss has not been experimentally shown to be crucial and therefore the technical novelty may be relatively limited, and 2) more comparisons are needed, to alternatives or other baselines.  I would be willing to change my decision on this, given supporting evidence.

**Experience Assessment:**

I have published one or two papers in this area.

**Review Assessment: Checking Correctness Of Derivations And Theory:**

I assessed the sensibility of the derivations and theory.

**Review Assessment: Checking Correctness Of Experiments:**

I carefully checked the experiments.

**Review Assessment: Thoroughness In Paper Reading:**

I read the paper thoroughly.

---

> ### Author Response · Authors · 2019-11-11
> **Reply to reviewer**
>
> 	* Technical novelty: while they address a problem that has not yet seen much attention, their solution is combination existing solutions. I say this because I am not fully convinced that the more novel aspects of their distillation procedure (i.e. the auxiliary task) are absolutely necessary. I would be willing to change my stance on this, given evidence.
>
> We agree that an ablation studies on the benefit of the auxiliary task is useful. However, we do not claim that the auxiliary task is the novelty of our work. In fact, we think training with auxiliary task is quite straightforward.
>
> The main contribution of our work is to draw attention to the meta RL problem in the batch setting by demonstrating that it is actually possible!
>
> 	* Limited comparisons to relevant alternatives, simple baselines.	 Do not compare to anything other than SAC.
>
> We compared against PEARL on the toy problem and PEARL did not manage to find the optimal policy, unlike our approach. This is mentioned in section 5.1. We do not highlight this fact enough in the current version of the paper and will make it more prominent in the revised version.
>
> 	* Only tasks considered are pointmass, hopper, and half-cheetah; other work (i.e. PEARL) has also been demonstrated on Ant in Mujoco.
>
> We are running experiments on Ant.
>
> 	* Argues that it is more stable than variants that involve interaction with the environment / use critic loss for MDP identification, which is a somewhat unfair since in their case 1) they assume they have data that is good enough to learn a good policy in the batch-RL setting, and 2) the stability is by virtue of the fact that they do things in a stage-wise manner because batch-RL works, i.e. they are doing supervised learning without having to bootstrap.
>
> (1) You’re right. We will highlight this limitation of our approach.
> (2) We are not sure why you think this is unfair. Actually, we argue that the stage-wise training pipeline is the strength of our approach.
>
> 	* Not sure if the use of term “pre-train” is appropriate, insofar as the test tasks are assumed to come from the same distribution as training tasks. It seems to be more about Batch-Meta-RL.
>
> You’re right. We will rewrite the paper to not use the term “pre-train”.
>
> 	* Not much attention given to attempts to solve the MDP mis-identification problem with simple solutions like giving side-information at training time (i.e. task-ID).
>
> We are not sure what you mean by this. The task-ID is not available in closed-form. The network has to learn to infer the task ID given the transitions from the task.
>
> 	* How much data per task is needed?
>
> For hopper, each task has 1 million transitions. For halfcheetah, each task has 60k transitions.
>
> 	* Harder tasks, where the method can’t approach SAC (as one would expect for sufficiently challenging tasks)
>
> We are running experiments on Ant.
>
> 	* Claim zero-shot generalization but in this case they study tasks where resets are not crucial for exploration needed for task inference
>
> We agree that our method has only been demonstrated to work on environment where exploration is not crucial for task inference. We will highlight this limitation in the revised paper.

---

> > ### Comment · AnonReviewer2 · 2019-11-15
> > **Further questions/response**
> >
> > 1. ** "We agree that an ablation studies on the benefit of the auxiliary task is useful. However, we do not claim that the auxiliary task is the novelty of our work. In fact, we think training with auxiliary task is quite straightforward." **
> >
> > In my original review, I noted:
> > "In all tasks considered, the transition function T does not change. Therefore, the function composite P(E(s_j, a_j)) should actually not learn any task-specific information when predicting s’_j. If the method works without the auxiliary loss, it is very similar to a stagewise version of PEARL, with Batch-RL."
> >
> > Am I right in my interpretation? As I said, if not for the auxiliary task, the presented method is very similar to a stage-wise version of PEARL, and without a stochastic model. If, as you say, the conception of the auxiliary task is "rather straightforward", then I feel this supports my case that the novelty is more limited than it may seem. In part due to the next point (2).
> >
> > 2.**"We are not sure why you think this is unfair. Actually, we argue that the stage-wise training pipeline is the strength of our approach."**
> >
> > Indeed, the stage-wise aspect is central to this work. I say it's unfair because you assume you have a replay buffer with good enough data of transitions from the environment that BatchRL works; other methods like PEARL, which are not stagewise, jointly solve the exploration problem, to eventually collect the requisite data. So the stage-wise approach in part because the setting considered is easier. Would the stage-wise approach work if considering data collected by a random policy? If so, this should be made more clear.
> >
> > 3.**"We are not sure what you mean by this. The task-ID is not available in closed-form. The network has to learn to infer the task ID given the transitions from the task."**
> >
> > Presumably you are given lets say k samples from the task distribution at training time; you use these training tasks in your batch-RL procedure to train specialized policies. Hence, you have the ground truth task ID at training time. My suggestion is merely to train a classifier that produces a posterior of training task IDs; and then condition a baseline policy on this training task ID. Not necessary, but certainly possible.

---

> > > ### Author Response · Authors · 2019-11-15
> > > **Reply**
> > >
> > > Thank you for your comments. We are here for a scientific discussion and your feedback is helpful in revising the paper.
> > >
> > > 1. Am I right in my interpretation? As I said, if not for the auxiliary task, the presented method is very similar to a stage-wise version of PEARL.
> > >
> > > You are right the interpretation that the proposed method is similar to a stage-wise version of PEARL. However, we are not after novelty for novelty sake only since the proposed method already performs well on standard benchmarks, matching the performance of SAC trained with 1 million samples in the case of hopper. We agree that testing the method on more complicated benchmark is interesting. We also agree that there exists with high probability approaches more sophisticated than introduced in our paper.
> > >
> > > But we feel this should be left for future work. The main point of this paper is to direct the attention of the community to meta RL methods that learn from observational data by formulating it and demonstrating that it is possible.
> > >
> > > 2.1 We do not agree that the setting we consider is easier because:
> > >
> > > - In meta-test, PEARL uses at least two exploratory trajectories, where the trajectory returns do not count towards the meta-test performance. We did not use such exploratory trajectories in our setting. Thus, PEARL setting is 2-shot, while our approach is zero-shot, where k-shot means that k exploratory trajectories are used during meta-test.
> > >
> > > - We tried PEARL on the toy setting, and it did not find the optimal policy, while our approach did. If our setting is really an easier setting than the setting in the PEARL paper, then PEARL should have found the optimal policy in the toy setting.
> > >
> > > - The settings that the PEARL paper considers and we consider are solving different problems. We do not argue that one is easier than the other (we do not argue that our setting is harder that the setting that requires exploration during meta-train, just different).
> > >
> > > 2.2 We agree that we need replay buffer with good enough data of transitions from the environment that BatchRL works. However, we argue that this is okay because:
> > >
> > > - We do not use any data from the meta-test MDP during meta-train.
> > >
> > > - It is reasonable to say that good enough data from the meta-train MDPs is a necessary assumption for any meta RL methods from observational data to generalize to the meta-test MDPs. If you agree that the setting of meta RL methods from observational data is interesting, then having this assumption is not a red flag.
> > >
> > > - The data in our replay buffer contain trajectories of varying qualities, some of which were generated by random policies.
> > >
> > > - In fact, even data generated by expert (i.e. human demonstrator) is straight-forward to use during the meta-train phase of our approach, whereas it is unclear PEARL can make use of such data, being a model-free RL algorithm.
> > >
> > > 3. Since the meta-test MDPs are not one of the meta-train MDPs, we are unclear what hypothesis your suggestion is testing.
> > >
> > > Also, what do you mean by the ground truth task ID at meta-train time? What would the representation of that be? In meta-train, we only have the different replay buffers and we do not know the MDP that generated each buffer. We do not have a one-hot encoding of the task ID. We do not know the transition and reward function of each meta-train MDPs.
> > >
> > > Would you please clarify if possible? How would the baseline policy make use of the task ID?

---

> > > > ### Comment · AnonReviewer2 · 2019-11-15
> > > > **reviewer response**
> > > >
> > > > 1. " The main point of this paper is to direct the attention of the community to meta RL methods that learn from observational data by formulating it and demonstrating that it is possible."
> > > >
> > > > I agree that the results shown do corroborate this, but I don't think the paper currently pushes this narrative. Shifting towards that narrative would make the paper stronger, IMO.
> > > >
> > > > 2. "- The settings that the PEARL paper considers and we consider are solving different problems."
> > > >
> > > > Agreed. Not sure I agree that this setting is not easier because perhaps being able to do zero-shot task inference / execution is by virtue of having 1) clean data and 2) tasks where task inference is not that hard / you don't need resets.
> > > >
> > > > 3. Can you please tell me what the auxiliary task of forward modeling does, or ablate it? It will not help for any task inference procedure because the transition function never changes!
> > > >
> > > > 4. "In fact, even data generated by expert (i.e. human demonstrator) is straight-forward to use during the meta-train phase of our approach, whereas it is unclear PEARL can make use of such data, being a model-free RL algorithm. "
> > > >
> > > > This should be fine for PEARL right? It's off-policy.
> > > >
> > > > 5. "Also, what do you mean by the ground truth task ID at meta-train time? "
> > > >
> > > > I simply mean that every time you get a replay buffer, you know the task ID. Just enumerate them.
> > > >
> > > >
> > > > ---
> > > >
> > > > Overall, this paper addresses a slightly different problem than the general Meta-RL problem, in the sense that it assumes sufficiently high-quality observational data and task identity at training time. I do think there is merit to this work, as long as the narrative presented is shifted towards something along the lines of "a simple stage-wise approach for meta-RL from observational data" or "distilling many policies into one for related tasks".
> > > >
> > > > Assuming this reframing happens, I'd be willing to change my review to (6) Weak Accept.

---

> > > > > ### Author Response · Authors · 2019-11-15
> > > > > **General Reply**
> > > > >
> > > > > Thank you again for your comments, which made clear you read the paper carefully and thought about it. As you can see, the other two reviewers are not as helpful as you are and your comments convince us there is still merit in the reviewing process to push science forward.
> > > > >
> > > > > We have reframed the paper in the direction that you suggested. Some of the changes include:
> > > > >
> > > > > - Changing the title of the paper to "Meta Reinforcement Learning from observational data".
> > > > >
> > > > > - Highlighting the desire to direct the attention of the community to this problem in the abstract.
> > > > >
> > > > > - Highlighting the desire to direct the attention of the community to this problem and the stage-wise nature of the approach at the end of the Introduction.
> > > > >
> > > > > - Mentioning the stage-wise nature of the approach and the distillation at the end of section 4.1.
> > > > >
> > > > > - Including the limitations of the current approach at the end of section 5.2.
> > > > >
> > > > > Considering the other two reviewers did not engage with us during the rebuttal period, we promise to include all the experiments you ask for if you fight for the paper during the discussion with AC, including:
> > > > >
> > > > > - Ablation of the auxiliary prediction tasks.
> > > > > - The classification baseline which enumerates the task ID.
> > > > > - Experiments on more challenging environments, such as Ant and Humanoid.
> > > > >
> > > > > We had started these experiments but were not able to finish them by the rebuttal deadline due to computational constraints.
> > > > >
> > > > > In case you think of more experiments you think we should try, we are also happy to make changes and add experiments you request after the rebuttal period is over in the final version of the paper if the paper is accepted.
> > > > >
> > > > > Please find below reply to your more specific questions:

---

> > > > > > ### Author Response · Authors · 2019-11-15
> > > > > > **More detailed reply**
> > > > > >
> > > > > > 2. …
> > > > > >
> > > > > > We agree that this approach was only tested in environments where reset is not needed for task inference. We have mentioned this limitation at the end of section 5.2. We will be happy to expand the scope of the experiments and test the approach on environments where reset are necessary for task inference, such as the sparse 2D navigation environment in the PEARL paper, if the paper is accepted.
> > > > > >
> > > > > > 3. … It will not help for any task inference procedure because the transition function never changes!
> > > > > >
> > > > > > You are right that the transition function never changes, so predicting the next state s' is unlikely to bring additional benefit. In the description of the approach in Section 4,  we mention predicting s' because we wanted to describe the general case of the approach.
> > > > > >
> > > > > > In the experiments already presented in the paper, the network only predicts r and does not predict s'. Since the reward function changes across the different MDPs, we argue that this makes sense. We are sorry we did not make this clear in the original submission and have revised the paper to mention this (before Section 5.1).
> > > > > >
> > > > > > 4. … This should be fine for PEARL right? It's off-policy.
> > > > > >
> > > > > > Actually, we don't know if PEARL will work without modifications if given expert data because:
> > > > > >
> > > > > > - PEARL uses SAC, one of the modern Deep RL algorithms, as the underlying RL algorithm.
> > > > > > - The Fujimoto paper (BCQ) shows that modern Deep RL algorithms diverge without modification when trained with transitions not generated by the policies that are being trained. What they really show is that there is different degree of “off-policyness” and current off-policy model-free RL algorithms do not work in the extreme off-policy case, where the transitions in the buffer are not generated by the policies that are being optimized for.
> > > > > >
> > > > > > Granted, The Fujimoto paper only tests on DDPG and DQN. But they show that the min of two Q function trick is not enough to solve the extrapolation error in value function. Since SAC only uses the min of two Q function trick and nothing else to combat the extrapolation error, we are uncertain whether PEARL will work without modifications when trained on data not generated by the policies that are being optimized for.
> > > > > >
> > > > > > That being said, given the results in PEARL and the Fujimoto paper, we agree that techniques from the Fujimoto paper can be adapted to PEARL with little effort and changes to PEARL so that PEARL works with expert data.
> > > > > >
> > > > > > To emphasize, we are NOT arguing that PEARL will not work. We are saying that we don’t know if PEARL will work and even if it does not, it can be made to combat the extrapolation error and train successfully with expert data in the standard Meta RL setting.
> > > > > >
> > > > > > On the other hand, we are arguing that PEARL without modification does not work in the Meta RL from observational data setting. We tried a PEARL-like approach on the toy experiment and found that it did not manage to successfully identify the MDP identity during meta-test. We even tried tuning different hyper-parameters of PEARL to try to make it work, such as the beta-weight of the KL term, or turning the probabilistic task inference function deterministic. This result was our motivation for the stage-wise approach. We have highlighted this in section 5.1 in the paragraph that starts with “Batch SAC with task inference function”.
> > > > > >
> > > > > > 5. "Also, what do you mean by the ground truth task ID at meta-train time? "
> > > > > >
> > > > > > Thank you for the clarification. It is now clear to us what the experiment you suggest is. We think the result of this experiment is more of a statement of the quality of the benchmark rather than the quality of any specific algorithm.
> > > > > >
> > > > > > That being said, as mentioned at the beginning of this message, we are happy to include this and other additional result in the final pdf if you fight for the paper during the discussion with AC and the paper is accepted.

---

### Official Review · AnonReviewer1 · 2019-10-23
**Official Blind Review #1**

**Rating:** 1

**Review:**

The paper focuses on a very interesting problem, that of pre-training deep RL solutions from observational data.   The specific angle selected for tackling this problem is through meta-learning, where a set of Q-functions / policies are learned during pre-training, and during testing the network identifies the training set MDP matching the data to extract a transferable solution.

The main strength of the paper is to draw attention to the issue of pre-training in RL, which is much less studied than in supervised learning, where it has been shown to have tremendous impact.   The paper also provides reasonable coverage of a large amount of related work.

Unfortunately I really struggled (despite careful reading) to understand several aspects of the proposed methods.   The training of function f() is not clearly explained; is this done as per Sec.2.4?  What is the loss function for this?  Is it done end-to-end as per the pipeline in Fig.1 (right), so using a gradient propagated back from Q?   What is the purpose of Proposition 1?  The more interesting point seems to be that solutions can “converge to a degenerate solution”, but this is not formally defined (i.e. how do you assess degeneracy, and how is that information used?)  Furthermore Proposition 1 seems limited to discrete state/action spaces. Is this the case for TIME in general?  The results on Mujoco suggest not.  Regarding the second phase of the pipeline, it is briefly mentioned that “P has low capacity” (bottom of p.4), but this is not explained further.  Is this due to a generalization issue, or a computational issue?  Why would P be low capacity but not E?  How does this actually impact the implementation?

As a higher-level comment: is it really necessary (preferable) to infer the identity of a specific train MDP (using the function f)?  This is used as a premise in this work, but I am not convinced this is desirable (for good generalization) or scalable (in the case of several observed meta-train MDPs).   What is the advantage of proceeding in this way?  Much of the work on pre-training in supervised learning just exposes the learner to large amounts of observational data to pre-condition the solution.

Finally, I have some concerns with the results as presented in the paper.  There are some details lacking, for example how specifically are the meta-test MDPs chosen for the Mujoco experiments?  How similar/different from the meta-train MDPs?  This is an issue because in Sec.5.1 the meta-test MDPs are chosen to coincide with meta-train MDPs.  So I am definitely interested in seeing how well the method actually generalizes to unseen MDPs, so need more detail on this part of the experiment.   I would also like to see a few additional naïve baselines.  First, what is the result if you do the pre-training as specified, and then at test time you randomly sample one of the pre-trained MDPs (rather than use the identification function).  Second, what is the result if you put all the meta-train data into a single batch, train a solution with SAC, then use this as a pre-trained solution (rather than the current “SAC trained from scratch”), allowing more training at test time.   Both these are useful sanity checks to verify the effectiveness of the proposed approach.

============
Post-rebuttal comments:

1.  My question, as stated in the review is: " how specifically are the meta-test MDPs chosen for the Mujoco experiments".  I read that they are tested on unseen MDPs.  I want to know how those unseen MDPs are selected / specified, and again as per my review: "How similar/different from the meta-train MDPs?"

2.  You should entertain the possibility that perhaps Sec.3 is not as clear as you think it is.  For example the MDP (S,A,\hat{T}, \hat{R}, \gama) is not defined as "the wrong MDP" - which I assume is what you mean by degenerate solution?

3.  I would like to know what are the results from the 2 naive baselines I described, as good sanity check.

In general, the rebuttal is intended to be a conversation to clarify understanding of the work. It is insulting to the reviewer to say that they did not read the paper carefully when they indicate they did.  It is much more productive to assume that many of your other (future) readers might have the same need for clarification, and you should be thankful for help provided by the reviewers to achieve this.

**Experience Assessment:**

I have published in this field for several years.

**Review Assessment: Checking Correctness Of Derivations And Theory:**

N/A

**Review Assessment: Checking Correctness Of Experiments:**

I carefully checked the experiments.

**Review Assessment: Thoroughness In Paper Reading:**

I read the paper thoroughly.

---

> ### Author Response · Authors · 2019-11-06
> **Reply**
>
> We do not believe your claim that you read the paper carefully because:
>
> 1. You mentioned that “I am definitely interested in seeing how well the method actually generalizes to unseen MDPs, so need more detail on this part of the experiment.”
>
> For the Mujoco environments, our approach is tested on unseen MDPs. This is mentioned at least 3 times in the paper: Figure 3 captions and the bottom of page 7 mention this in bolded text. This is also mentioned in the abstract.
>
> 2. You mentioned that “The more interesting point seems to be that solutions can “converge to a degenerate solution”, but this is not formally defined”.
>
> This is formalized in Proposition 1. The paragraph above Proposition 1 clearly reads “previously proposed solutions ... converge to a degenerate solution … We can formalize this statement with the following proposition.”
>
> The reviewer pose questions that are clearly answered in the paper. This convinced us that the reviewer did not read the paper carefully as the reviewer had claimed.

---

### Author Response · Authors · 2019-11-06
**General Reply and Comment on Reviewer Quality**

We call out reviewer 1, which claims to have read the paper carefully. However, the reviewer’s questions indicate the reviewer clearly did not (details mentioned in direct response to the reviewer).

We thank reviewer 2 for carefully reading the paper and asking good questions.

Reviewer 3 reads the paper hastily, as confirmed by the statement “I made a quick assessment of the paper”, and the reviews miss important points.

We understand that there are many papers on reviewer’s plate and it is the authors’ job to educate the reviewers. In the individual response to the reviewer, we will engage with good feedback, revise the papers where the reviewers found unclear and correct misunderstandings.

---

### Author Response · Authors · 2019-11-15
**End of rebuttal summary**

Reviewer 2 was a good reviewer, who engaged with us in multiple rounds of discussions. The reviewer promises to raise the score to 6 (weak accept) if we change the story as the reviewer suggests. We think we have done this in good faith.

We also promise in good faith to include in the final pdf all the experiments reviewer 2 suggests if the paper is accepted.

Reviewer 1 and 3 did not read the paper carefully and did not engage with us during the rebuttal period. We have tried to the best of our abilities to address their comments in the revised paper.

---

### Decision · Program_Chairs · 2019-12-19

**Decision:**

Reject

**Comment:**

The reviewers reached a unanimous consensus that the paper could not be accepted for publication in its current form. There were a number of concerns raised regarding (1) the clarity of the writing; (2) the comparisons, especially to prior work; (3) the details of the experimental setup.